# Robustness of Deep-Learning-Based RF UAV Detectors

**DOI:** 10.3390/s24227339

**Published:** 2024-11-17

**Authors:** Hilal Elyousseph, Majid Altamimi

**Affiliations:** Electrical Engineering Department, College of Engineering, King Saud University, Riyadh 12372, Saudi Arabia; elyousseph.4@gmail.com

**Keywords:** UAV detection, signal processing, spectrum monitoring, computer vision, deep learning

## Abstract

The proliferation of low-cost, small radar cross-section UAVs (unmanned aerial vehicles) necessitates innovative solutions for countering them. Since these UAVs typically operate with a radio control link, a promising defense technique involves passive scanning of the radio frequency (RF) spectrum to detect UAV control signals. This approach is enhanced when integrated with machine-learning (ML) and deep-learning (DL) methods. Currently, this field is actively researched, with various studies proposing different ML/DL architectures competing for optimal accuracy. However, there is a notable gap regarding robustness, which refers to a UAV detector’s ability to maintain high accuracy across diverse scenarios, rather than excelling in just one specific test scenario and failing in others. This aspect is critical, as inaccuracies in UAV detection could lead to severe consequences. In this work, we introduce a new dataset specifically designed to test for robustness. Instead of the existing approach of extracting the test data from the same pool as the training data, we allowed for multiple categories of test data based on channel conditions. Utilizing existing UAV detectors, we found that although coefficient classifiers have outperformed CNNs in previous works, our findings indicate that image classifiers exhibit approximately 40% greater robustness than coefficient classifiers under low signal-to-noise ratio (SNR) conditions. Specifically, the CNN classifier demonstrated sustained accuracy in various RF channel conditions not included in the training set, whereas the coefficient classifier exhibited partial or complete failure depending on channel characteristics.

## 1. Introduction

Recently, unmanned aerial vehicles (UAVs), also known as drones, have seen an upsurge in usage. This can be contributed to both positive and negative developments. On the one hand, UAVs have been used for surveying, power line inspections, agriculture, rescue missions, etc. On the other hand, inexpensive commercial off-the-shelf (COTS) UAVs have also been used for terrorist attacks against civilian infrastructure such as oil facilities and airports [1,2]. Since most UAVs will rely on multiple radio frequency (RF) links, this work investigates how RF spectrum sensing, in combination with cutting edge machine-learning (ML) and deep-learning (DL) techniques, can be used for UAV detection. Utilizing ML and/or DL techniques to classify RF waveforms has been developed more broadly, and is by no means specific to UAV waveforms. For example, it has been used for classification of radar waveforms [3,4,5], communication protocols [6,7,8], and automatic modulation classification [9,10,11]. It has also been used as an enabler for cognitive radio [12,13,14]. Although applying these techniques specifically for UAV detection is a subset of the general RF ML/DL based signal classification, there are still lessons that can be learned from the general RF signal classification literature, such as how to preprocess the RF signals. In fact, even general RF signal classification has taken advantage of developments in the overarching field of DL-based classification, evident by researchers using models for RF signal classification that were originally developed for the computer vision field [15]. In order to not extend the discussion too broadly, we have chosen to focus this work on UAV detection.

While UAV detection can be performed by other means, such as radar, cameras trained to perform object recognition, or microphone arrays trained to recognize the sound of a UAV, all methods have trade-offs to consider. Radar-based UAV detectors are an excellent choice for two main reasons. Firstly, they are able to detect a UAV that is not utilizing any RF link. The second reason is that radar-based methods tend to have the largest detection range. Nonetheless, radar-based detection suffers a major disadvantage, which is that it requires active transmissions. To mitigate this risk, an alternative is to use passive radars. Passive radars require taking advantage of an existing signal of opportunity in the environment. If the signal of opportunity is lost, the method will not work. Camera-based methods, whether optical cameras or infrared (IR) cameras, are also a promising method. Similar to radar-based methods, camera-based methods can also detect UAVs that are not using RF transmissions. Camera-based methods achieve this benefit, but without the risks or costs associated with radar. There are however two major downsides of camera-based methods. The first disadvantage is that they can confuse other objects with UAVs, such as birds. The second disadvantage is the low detection range, which is worse than radar- and RF-based methods. Camera-based methods can only detect the UAV as far as the optics allow, and also only in the direction that the optics are pointed towards. The only detection methods with a shorter range than camera-based methods are acoustic-based methods, which are the least popular of all methods. Acoustic methods utilize microphone arrays, which can provide both detection and geolocation. However, the major disadvantage of acoustic methods is that they have the shortest detection range of all methods.

In order to gain insight as to which existing UAV detection methods are gaining the most traction, we can consider the work of [1], which performed a market study. The study tallied 537 counter-UAV systems available on the market, with a breakdown by percentage shown in Figure 1. The study showed that RF-based systems are the most popular method to detect for the presence of UAVs. This is to be expected as almost all UAVs will be using at least one RF link, whether for control, telemetry, or video down-link [16]. RF-based methods do not strictly imply the inclusion of ML/DL techniques, which are still being actively developed. An example of an RF-based technique without ML/DL is to simply demodulate the telemetry packets [17], which could include information containing unique identifiers. This depends on a priori knowledge of the target UAV, which might not be available. Therefore, the recent trend is to augment RF-based methods with ML and DL techniques.

Despite recent successes with UAV RF ML/DL-based detection research, there is a major gap, which is the robustness of such models. It is not clear whether many of the proposed methods are building truly robust UAV detectors or if they will perform optimally against a specific dataset. Many different approaches compete against public UAV datasets [18,19,20], sometimes achieving accuracy up to 99% [21,22], which is very promising. However, one issue is that the public datasets are usually collected in a lab setting, with simulated noise added later, which does not accurately capture real channel conditions. A second issue is the test data are simply extracted from the training data, meaning the RF recordings are from one setting. This introduces a major risk, which is that the UAV detectors may be reporting very high accuracy scores, but could still fail when deployed. We will refer to this parameter of a UAV detector as the robustness of such a detector. Testing the robustness requires a different approach than used thus far. The current literature has seen different proposals on how to achieve optimal accuracy against a specific dataset. Given that the test sets have been extracted from the training pool, there is reason to believe that even detectors reporting near 100% accuracy may fail in a different RF environment. This could be due to new RF transmissions that were not found in the training set. It could also be due to the same RF transmissions, but with different signatures, such as the power levels, frequency channel occupied, or the general congestion of the channel. Therefore, it is necessary to extend the datasets by adding more recordings from different settings and locations, to better capture real-world channel conditions and background RF activities. To be clear, the required extension is in terms of quality of the data as opposed to the sheer quantity. In terms of training data quantity, it has already been shown in [23] that training saturation begins in as little as 50 training samples per class, where the authors found negligible accuracy improvements after that point. Therefore, the extension required is to repeat the physical setup of existing works, but to capture test data from different channel conditions and RF background activity patterns. This is the only way to ensure that a model that may score 100% accuracy in one test location will still perform well in a different scenario. Unfortunately, the public UAV datasets are difficult to extend in this manner, namely due to:Utilizing multiple high-end SDRs [19].Expensive lab grade spectrum analyzers [18].Utilizing proprietary licensed software (such as LabVIEW or MATLAB) [18,19,20].Signal data are already pre-processed into a certain signal format [20].

To emphasize, the gap is that it is difficult to decipher if UAV detection methods proposed will truly be robust to different real-world conditions. The high-entry barrier, on the other hand, is considered an obstacle towards addressing this gap. This is because addressing this gap entails extending the test datasets. Therefore, we found it necessary to introduce a new UAV dataset with an easier entry-barrier. To this end, we utilize an SDR and develop all experiments in Python, which is open source and easy to use. Data are collected by physically varying the locations and the received SNR, as opposed to adding simulated noise. The received SNR is varied by adjusting the distance between the SDR and the UAV controller. Data were collected over a period of several days and in several locations. In total, there are 1100 RF recordings, capturing different RF activities and SNR conditions. Since only COTS devices and open-source software are used, the dataset is easily extendable to add new UAV data or produce further robustness tests.

The organization of the rest of the paper is as follows. In Section 2, we review different open source UAV datasets and the performance of different approaches, while discussing the shortcomings. In Section 3, we introduce our methodology. In Section 4, we introduce the setup of our experiments. In Section 5, we provide the results and analysis, before concluding with Section 6.

## 2. Literature Review

To study the existing UAV RF ML/DL-based detection works, it is helpful to organize the discussion based on the dataset used, followed by the approaches attempted on each dataset. The first UAV public dataset is published in [19]. The authors used three different UAVs, being the Bepop, AR, and Phantom UAV models. They used National Instruments USRP receivers, each with a maximum bandwidth of 40 MHz, and LabVIEW software to record the signals. To cover 80 MHz of the ISM band, they utilize two receivers and stitch the results together. The authors recorded anywhere from 5–10 s of different RF activities. As a first step, they recorded whether a UAV was active or not. In other words, all UAV signals became one class, and the classifier was trained on a binary decision on whether a UAV was present or not. The next classification task was divided into four classes, to include “UAV off”, Bepop UAV, AR UAV, or Phantom UAV. Finally, they extend to 10 classes in the third experiment by attempting to classify the flight mode of each UAV. The authors introduced a dense neural network (DNN) and train the models on the discrete time Fourier transform (DTFT) power values of the RF recordings in the frequency domain, as a 1-D array.

The dataset has been tested on by others [24,25,26,27,28,29,30], including our previous published work [31], where we achieved performance gains by introducing power spectra averaging before creating images and training an image-classifying CNN. The majority of the works have used CNNs for the classification, which outperformed the original DNN. The CNN approaches vary depending on the pre-processing steps and the actual CNN architecture used. All of the authors have stuck to frequency domain values as the input data, with the main difference being the ML/DL architecture, and whether classifiers were trained on numerical arrays (i.e., coefficients) or on images. The VGG16 Neural Network was trained on images in [30], which outperformed the CNNs for the four class scenario of [25,29], but not the 10 class scenario of those works, whom utilized arrays. The authors in [29] using coefficients scored about equal to our image-based approach, by using the XGBoost algorithm for the 2 and 4 class experiments, and outperformed our model for the 10 class experiment. Although the XGBoost coefficient approach outperformed our image classifier, it did not outperform the image classifier used by [30]. Therefore, an initial comparison of these approaches is largely inconclusive. Some general trends hold true, such as that CNNs and XGBoost have performed the best, yet one approach has outperformed the other depending on the classification experiment. The dataset is not broken down into signal to noise (SNR) categories, which is one reason why it is difficult to conclude which approach is optimal. It is likely that approaches can perform identically with high SNR, but only the truly robust approach will maintain accuracy under low SNR conditions.

The newest work we have found on this dataset is that of [28], which was able to score 99% for the 10-class scenario using hierarchical learning, which was a significant boost compared to all other approaches for the 10-class scenario. Almost all approaches have already achieved 99% accuracy for the 2-class case (binary detection), so the goal has been to find a model that boosts the 10 class performance. The approach of [28] essentially divided up the 10 classes into three different stages with three different models. For example, one model was specialized in a binary UAV detection. If a UAV was detected, another model would be called, specialized in determining the UAV type. Only after the UAV type is determined would a third model be called to predict the flight mode of the specific UAV. Therefore, the 10-class scenario is essentially broken down into multiple 2- or 3-class scenarios. While we can learn from this work that accuracy can be improved by reducing the number of classes each individual model has to train for, it is not clear if the end result is simply an optimized way to master the specific dataset. As a reminder, each class was taken all in one setting, and from one location. Therefore, the test data are extracted from the training data prior to training. Although in principle this may be fine, it is not clear how each approach would work against test data from different locations and signal strengths. We show in the results section that different approaches may perform on par in the same scenario, but then have significant degradations in a new scenario. To test for this, the same UAV signals could be again recorded in new settings and added to the dataset, although this would require two USRP-2943R receivers, PCIe interfaces, and LabVIEW Communications System Design Suite.

Another popular UAV dataset has been published in [18]. The dataset extends from 10 classes to 17 classes by introducing different UAV controllers. The dataset is taken outdoors as opposed to the previous set. An expensive oscilloscope with a sampling frequency of 20 Gigasamples per second is used. The overall approach is quite different to the previous dataset. Instead of having a “UAV off” category, the authors perform energy detection, and only attempt classification if a threshold is exceeded. Different ML classifiers are used to distinguish different UAV controllers. The authors use MATLAB to add noise to the signals and test the performance in different SNR regions. The authors mention that training the models took several hours for all 17 classes. The best performing classifier was random forest, achieving upwards of 98% accuracy. The work relies on manual extraction of expert features, and works on classifying arrays. The authors update their work in [23], where they switch to spectrograms and image classifying CNNs instead, achieving superior results for lower SNR values. This again emphasizes an important note, which is that we have found different approaches perform almost identical for high SNR values, whereas gauging the best approach requires testing under different SNR conditions. The additional benefit of using the CNNs was to remove the need for feature extraction. It should be noted that although the CNN had superior performance even under low SNR conditions, the authors introduce a pre-processing step that de-noises the signal before attempting classification. The authors also attempt time-series images for the classification, and found that the time-series images had an accuracy of about 50.1% for 0 dB SNR, while the spectrogram images had an accuracy of 82.9% for the same SNR. The authors also introduce the concept of judging classification accuracy by training set size. The authors add simulated noise in the range of −10 to 30 dB in steps of 5 dB. Breaking down different SNR steps as classes, the authors show that with as little as 20 samples per class, equally distributed to the different SNR classes, the CNN models perform quite well. Accuracy eventually “saturates” after 50 samples per SNR class, meaning the authors stopped seeing accuracy gains for the increased training data.

The newest work we found using the previous dataset was [32]. The authors introduce a transformer model named “SignalFormer”. Contrary to previous CNN approaches, the authors do not rely on simply providing images to the CNN. The authors introduce feature extraction procedures that aim to capture frequency and time axis dependencies. At the cost of increased complexity, the authors outperform the accuracy of [23]. The authors measure the increased complexity in FLOPS (floating point operations) of the model, where we note that this does not necessarily capture the time introduced by pre-processing steps. It does however help to indicate how much time the ML/DL inference will consume, relative to the compute platform. The strong point of the research is that the authors achieve 93% accuracy under SNR conditions as low as −15 dB. Although this dataset has less competing author approaches attached to it, some general trends hold. Namely, researchers have tended towards CNNs, and have found that adding pre-processing steps can increase accuracy performance. Increasing the accuracy is certainly a welcome contribution and an excellent first step. Our claim however, is that, similar to the previous dataset [19], comparing different approaches as they optimize for accuracy does not give a complete picture. It is still necessary to test the different approaches to see which approach will not only have the best accuracy on test data from the same scenario, but also to see which approach performs accurately on data taken from different environments or scenarios. Similar to the first dataset, the test data in this experiment is also extracted from the same recording as the training data.

A third UAV dataset [20] uses a USRP with a sampling rate of 100 MHz, and captures the RF spectrum of 9 different UAV controllers and also a Wi-Fi router inside an anechoic chamber. After inspecting each recording, they add simulated noise to the samples using MATLAB. The authors then converted the signals into spectrogram images, which are used for the training and test datasets. In general, the results are very good and can reach up to 99% for low SNR values using an image classifying CNN. There are less follow-up works for the above dataset, possibly because the authors have already achieved 99% accuracy even at 0 dB SNR, or because the shared dataset is already in the form of spectrogram images (limiting the amount of different processing steps that could be applied). A third reason could be the setup requiring a high-end USRP X310 (lab-grade SDR model) and an anechoic chamber, which would make extending the work difficult.

An analysis of the existing UAV detection literature concludes with some important notes. It is certainly good news that UAV detectors have achieved above 99% accuracy. Perfect accuracy in a specific lab setting, however, does not necessitate successful real-world performance. In our analysis, we have found that although there are many proposed UAV detectors, they are all working towards the same objective, which is to master a specific dataset. Moreover, as many proposed models achieve equal accuracy, we do not obtain a good indication as to which model to utilize. This gap is compounded upon by the fact that closed source tools and expensive lab-grade grade equipment have been used to capture the data. Although the dataset itself is published openly, extending the dataset to test for robustness will be difficult. In this work, we aim to provide an easy to extend research by introducing the first UAV dataset taken with a COTS SDR. Instead of using two SDRs or lab-grade equipment with extremely high sample rates, we introduce sequential scanning, which captures the same amount of RF spectrum using rapid snapshots. We access the SDR with the Python programming language, which is open-source, so a license is no longer required such as for LabVIEW or MATLAB. The previous works have achieved high accuracy, but unfortunately, there remains the open question with regards to robustness. By robustness, we are referring to the ability of a model to maintain suitable performance in new channel conditions not included in the training dataset. Testing robustness requires an extendable test UAV dataset, along with a methodology that is easy to replicate, which we will introduce next.

## 3. Methodology

In this section, we discuss in depth the overall methodology of our work. This includes introducing the data format and data collection procedures. Before introducing the data format, we briefly introduce the overall block diagram of a ML/DL RF-based UAV detector in Figure 2.

Starting from the left-most side, we notice there are many devices that can transmit RF signals. In this case, we are interested in the UAV signals, which could be either the uplink control signal, or downlink telemetry/video. These signals can be captured by a receiver, such as an SDR, with the signals viewed on a computer program which displays energy/power for different RF frequencies. Unfortunately, detecting a UAV by means simple energy detection threshold is not feasible, as there will be many false results from Wi-Fi or other ISM devices making RF transmissions. This motivates the use of DL techniques, which will learn to output an alert only if the signature of the UAV signal is detected. In order to utilize DL techniques, a good understanding of the data used is first required, which will be discussed next.

### 3.1. Data Format

In this work, a COTS SDR, HackRF One, was used to sample the RF spectrum. The HackRF One SDR has a maximum sampling rate *Fs* of 20 million samples per second (Msps). These sample points are grouped together as a complex valued sample, which is referred to as an IQ sample, where *I* is the in-phase coefficient and *Q* is the quadrature coefficient. The data format is shown in Equation (Equation 1), where *x*[*n*] represents one “IQ sample”.
(1)x[n]=I+jQ

For the identification of wireless signals, it is more advantageous to convert these discrete time domain values into *N* discrete frequency domain values by the DTFT (discrete time Fourier transform), by means of Equation (Equation 2), where *N* is the number of DTFT points.
(2)$$X\left[k\right]=\sum_{n=0}^{N-1}x\left[n\right]e^{-j\frac{2\pi kn}{N}}$$

The number *N* chosen dictates the number of output coefficients, *X*[*k*], produced. In these experiments, a value of *N* = 1024 is used. Thus, the time domain IQ samples of form *x*[*n*] became 1024 frequency domain values of form *X*[*k*]. Finally, it is more common to estimate the power spectrum density (PSD) of the frequency spectrum, by means of calculating the periodogram, shown in Equation (Equation 3).
(3)$$P\left[k\right]={\left|X[k]\right|}^{2}$$

In this research, the Python library Matplotlib was used to calculate the PSD. By default, it will apply a Hanning window to the time domain data, as well as performing averaging. This resulted in an array of 1024 real data points for each scan. In our experiments, as the maximum scan rate of IQ samples for our hardware setup is 20 Msps, we can capture 20 MHz of bandwidth of RF spectrum. To capture the entire 2.4 GHz ISM band, 2.40 GHz to 2.48 GHz, 80 MHz in total, we sequentially scan 20 MHz at a time, which results in four “snapshots”. We took the PSD of each snapshot before concatenating the results together. This resulted in a real array of 4096 values, which are saved as numpy arrays. These values are also plotted and saved as images. Turning on our generic UAV controller, we can see the UAV control signal isolated between the two black-dotted lines, shown in Figure 3.

In our experiments, there are two main data formats used. The first is the PSD of the RF samples in the form of a 1-D array, as explained above. The second format is to plot these values, resulting in an image, as was shown in Figure 3. A key objective of choosing these two different formats in this work is to compare whether better performance is achieved by treating the data as a 1-D array and applying coefficient classifier techniques, or to treat the data as an image and apply computer vision techniques. Now that we have discussed the data formats in depth, we will discuss the data collection procedure.

### 3.2. Data Collection

The timing of the experiment was across several days to collect all the data required. The RF signal recordings are taken in snapshots, with approximately 30 min–1 h required for each setup. This time includes setting up the equipment, adjusting for the correct distance, before finally initiating the Python scripts that will make the RF recordings. The actual RF recordings are several seconds for each train and test category, although we note that even 1 s of RF data consists of 20 million sample points. For comparison, the UAV dataset of [19] has 10.25 s of RF recordings, whereas our dataset has approximately 34 s of RF data. The setup to collect the data is shown in Figure 4. The SDR was connected to a Lenovo Laptop with an Intel Core i7-7500U CPU @ 2.70 GHz, 8 GB of RAM, and running an Ubuntu 20.4 operating system. Python scripts were developed to access the SDR and perform various functions. This includes, for example, tuning the SDR to the desired center frequency. After tuning, the SDR can be commanded to stream IQ samples. The IQ samples are converted into PSD values as described above. In summary, the data collection process can be conducted in five steps:

Sample the RF spectrum for a fixed number of samples, referred to as a time snapshot. We collected 625,000 IQ samples per snapshot, corresponding to a time snapshot of 0.03125 s.Perform signal processing steps on the data, such as de-averaging [33].Convert the time domain samples into *N* frequency domain power spectrum values via the PSD method.Save the *N* PSD values into a 1-D array and also as an image.Save the resulting arrays and images into folders.

The RF spectrum was scanned first for background activities with no UAV signal present, and then again with the UAV controller emitting. For the case of UAV signals, the labels include meta data indicating the location of the recording and the distance from the controller. For our experiments, we conduct the data collection in three RF environments and for several SNR readings. At the end of this process, we acquired a training dataset, and repeated for a separate test dataset. The procedures for collecting the training data and test data are identical. This concludes the methodology. In the next section, we will discuss the experimental setup, starting with the test dataset.

## 4. Experimental Setup

This section discusses the test dataset, the types of classifiers used, the hardware setup, and the actual experimental models.

### 4.1. Test Dataset

We can break down the test set, which is equally distributed into the following eight categories, with 100 recordings/images in each. The lowest SNR category is repeated three times, as the main goal of our work is to test low SNR performance.

RF background (Environment 1)RF background (Environment 2)UAV controller at high SNR (Environment 1)UAV controller at medium SNR (Environment 1)UAV controller at low SNR (Environment 1)UAV controller at lowest SNR (Environment 1)UAV controller at lowest SNR (Environment 2)UAV controller at lowest SNR (Environment 3)

Environment 1 was conducted in an indoor setting, Environment 2 was conducted in an outdoor setting, and Environment 3 was conducted in another outdoor setting but in different surroundings. For the indoor settings, we allowed multi-path obstacles to exist between the UAV controller and the SDR. Figure 5 shows example plots of the test data recorded at these locations.

In a simple manner, we perform the detection into two classes: “UAV present” versus “RF Background” (i.e., no UAV present). In the case of “UAV present”, the data themselves are further sub-divided by way of meta-data to perform different experiments and gain insights. Attempting to increase the maximum distance between the SDR and the UAV can be achieved by using directional antennas, or utilizing a stronger UAV control signal. Reporting detection distances in such a manner (i.e., directional antennas) causes the results to be strongly dependent on the test setup. Therefore, we used an omni-directional antenna, and allowed the UAV signals to be recorded in different SNR categories. We consider different SNR levels as: high, medium, low, and lowest. The lowest SNR category was achieved by decreasing the received power until the UAV signal was just slightly visible above the noise floor. The SNR was then increased to achieve the different categories. In this fashion, we are able to investigate the performance of the different classifiers in terms of received SNR. Optimizing classifier performance in terms of SNR will automatically translate into increasing the detection distance of a model in general terms, whereas the actual detection distance (in meters) is then purely a function of the receiving station gain and transmit power of the UAV. Next, we discuss types of classifiers used.

### 4.2. Classifiers

We utilized two main techniques in this work, the XGBoost algorithm to represent ML/coefficient classifiers, and a CNN to represent DL/computer vision classifiers.

#### 4.2.1. Image Classifier

Our main CNN neural architecture has already shown a good accuracy for up to 10 classes on the dataset of [19], as introduced in [31], but is shown again in Table 1 for completeness. As a recap, all convolutional layers had a filter size of 3 by 3. The models were implemented using TensorFlow/Keras. The Adam optimizer was used, and our models trained to minimize the categorical cross entropy. Training was performed for 50 epochs. Our model has around 900K trainable parameters, giving a memory footprint of about 11 MB. For reference, the popular VGG16 model has 138.3 M parameters and a size of 528 MB, meaning our model is significantly lighter.

#### 4.2.2. Coefficient Classifier

The XGBoost algorithm was chosen as our second main technique to represent coefficient-based algorithms. It is based on decision trees and is an improvement on the random forest and gradient boosting techniques. It has been shown in in previous UAV detection works that this algorithm has outperformed other ML classifiers [29,34,35]. It has even outperformed DL image-based classifiers for the open-source UAV dataset of [19], although the result is not fully conclusive, which is a motivation for this work.

Training the XGBoost model in this work was done with the XGBoost library in Python. Training was identical to our previous work [31] in which the algorithm was able outperform our CNN. In our previous work, XGBoost was able to achieve 93% accuracy, compared to our CNN which achieved 91.7% accuracy. For XGBoost, the original 1-D PSD coefficients are used, which are the same values plotted when training the image classifying CNN. We stored the coefficient values as numpy arrays. The different arrays (PSD coefficients of RF background, PSD coefficients of UAV signal present with high SNR, PSD coefficients of UAV signal present with low SNR, etc.) are loaded back into Python and then vertically stacked. Label arrays are created which are “RF Background”, and one label “UAV on” for all SNR categories. These are then split with 90% training and 10% validation using the Sci-kit learn library. This concludes the discussion on the classifiers used in this research. Next, we discuss the hardware setup.

### 4.3. Hardware Setup

In these experiments, an Ettus Research VERT2450 omni-directional antenna with a gain of 3 dBi was used. The HackRF SDR used has three gain stages in its receive architecture, and the values were set as low noise amplifier (LNA) gain = 16 dB, variable gain amplifier (VGA) gain = 16 dB, and final amplifier turned off. For perspective, the maximum LNA gain available is 40 dB, the maximum VGA gain available is 62 dB, and the amplifier, if enabled, provides an additional 14 dB of gain. As we kept the gain values fixed, these tests serve to investigate increasing the relative successful detection distance of any system in general, as opposed to researching how to achieve a maximum distance for a specific setup. With fixed hardware parameters, the overall workflow is thus to train different experimental models, and compare the performance of the different models against the test set.

### 4.4. Experimental Models

In these experiments, the training pool consisted of up to 300 signal recordings, which are identical between the different classifiers, other than one training set, which was saved as arrays, and the other set saved as images. Unlike the test set, the training set only has UAV signal recordings from Environment 1. This was done to increase the difficulty of the test set. The actual training data drawn from the pool define the specific experimental models used. The models are shown in Table 2:

## 5. Results

This section introduces the performance metrics, results of different experiments, and an in-depth interpretation of the results.

### 5.1. Performance Metrics

The main performance metric used thus far for UAV detection works has been accuracy. The accuracy of a model is defined as the number of true positives (*TP*) plus true negatives (*TN*), divided by *TP* + *TN* along with the false positives (*FP*) and false negatives (*FN*). It is formally defined in equation form below.
(4)Accuracy=TP+TN/(TP+TN+FP+FN)

It is also common to define the precision and sensitivity of a detector. A detector with high precision will yield low false positives, i.e., it will only alert for a UAV detection if one is truly present. Sensitivity on the other hand defines missed detections. A detector with high sensitivity will have a lower probability of missing a UAV detection, but it will be more likely to output false positives.
(5)Precision=TP/(TP+FP)
(6)Sensitivity=TP/(TP+FN)

We note that the precision score is penalized by false positives, whereas the sensitivity score is penalized by false negatives. The precision and sensitivity are usually combined into one metric, the *F*1 score, which is the weighted mean of both metrics. The *F*1 is used to mitigate class imbalances, i.e., if there are more possible instances of false positives versus false negatives. In our test set, there are 200 instances of RF background, and up to 600 instances of UAV signals. This means there are 200 possible instances of false positives, as opposed to 400 possible instances of false negatives. Therefore, the *F*1 has been reported along with the accuracy, although we will see the two metrics are in close agreement.
(7)F1=2×(Precision×Sensitivity)/(Precision+Sensitivity)

The novel contribution of this work investigates a new concept with regards to UAV detection, which is robustness. Robustness is a relative measure, and to the best of our knowledge it has not been formally defined thus far. A proposed definition would be to simply measure the accuracy degradation a model experiences after accounting for a specific weakness. In other words, the delta between the baseline accuracy *Ab* and the new “robustness accuracy” score *Ar*. We subtract the resulting delta from one, so that the maximum score becomes one instead of zero.
(8)R=1−(Ar−Ab)

A maximum robustness would be indicated by no accuracy degradation, for a score of 1, i.e., 100% robustness. If a model becomes 10% less accurate, the robustness score would be *R* = 90%, and we could say the model is 90% robust to the introduced weakness.

### 5.2. Baseline Accuracy Performance

Our top performing models are CNN-300 and XGB-300. These were trained with the full training set of 300 examples of RF background images and UAV signal images of different SNR categories, as outlined in Table 2. The results for accuracy and different detection metrics are presented in Table 3, followed by the accuracy breakdown for different categories in the test set in Table 4.

We can notice that XGB-300 slightly outperforms CNN-300 overall. CNN-300 misclassified two instances of Background 1 to be instances of a UAV present. These two false positives resulted in a slight reduction of the precision score. CNN-300 also had a slightly reduced sensitivity score, as it missed three instances of UAV signals.

XGB-300 had the same precision as CNN-300, but had a perfect sensitivity score, since it did not miss any UAV signals. These overall results have matched our previous publication [31], where XGB slightly outperformed our image classifying CNN for the public UAV dataset of [19]. As we did not find interpreting the precision, sensitivity, and *F*1 scores too relevant to the robustness discussion, the rest of the experiments focus on the accuracy score, which is the only metric needed to calculate robustness.

### 5.3. Robustness Performance

The accuracy performance presented above matches the results of UAV works in the literature thus far. However, when introducing robustness, we noticed the results changed. A weakness in UAV detection works specifically, and DL/ML-based signal recognition tasks generally, is performance under the lowest SNR conditions. Therefore, we tested the lowest SNR performance of the models with respect to different RF channels. We used three different “lowest SNR” recordings as introduced in Section 4.1. These categories were intended to maintain similar SNRs to each other but that are taken from different locations. Different locations will experience unique channel conditions, as well as different patterns of RF background activity, both of which will influence the performance of the models. In order to calculate the robustness, we must first calculate the accuracy performance in the new test categories. The results are shown in Table 5:

“UAV Lowest Environment 1” is the baseline accuracy for the lowest SNR category as was reported in Table 4. From the new test categories, we see that CNN-300 classified 81 and 53 UAV signals correctly, instead of 99 as it did in Environment 1. This is a degradation of 18 and 46 for Environments 2 and 3. Using the robustness equation, we can translate the degradation into robustness scores of 82% and 54%, for an average robustness of 68%. XGB-300 in Environment 2 successfully classified 44 UAV signals. In Environment 3 on the other hand, XGB-300 failed almost completely, with only one positive detection. This is an absolute reduction of 56 and 99, respectively. The robustness scores for XGB-300 are thus 44% and 1%, for an average robustness of 22.5%. Therefore, on average, CNN-300 is 45.5% more robust to low SNR conditions. This is displayed Figure 6 and summarized in Table 6.

Therefore, although the XGB coefficient classifier at first glance appears to have superior performance to the CNN image classifier, the results do not hold when replacing UAV test categories with UAV recordings from different RF channel conditions. Examining the test data of Environment 2 compared to Environment 3, which had very similar received UAV SNR, there is nothing that immediately strikes out as a reason for the degraded performance of XGB-300 compared to CNN-300. The only difference is that the two test categories were recorded in a different setting, and at a different time of the day. Aside from classical propagation topics, such as multi-path, frequency offset, etc., for our DL- and ML-based classifiers, we must also consider that the different environments will have a different pattern of RF background activities. This supports our suspicion that test data should be a separate recording, as opposed to extracting it from the training pool. In the literature, the test UAV data has been taken from the same pool as the training data, which is why it is difficult to assess how robust the proposed models truly are. It also supports our suspicion that a model could perform excellently under a certain condition, but then fail completely or partially in another condition. To further explain this behavior, we re-trained the XGB and CNN models while gradually incrementing their training data, as was outlined in Table 2.

### 5.4. Robustness Explanation

An explanation for the superior robustness of the image classifier is learned by repeating the experiments while gradually building up the training data. Starting “backwards”, we introduce models CNN-100 and XGB-100, which are trained with only 50 RF background examples and 50 UAV high SNR examples. We use these models against the entire test set. Both models score extremely well in predicting RF background as well as high SNR UAV images. The robustness realization happens when observing the performance results for the medium and low SNR images in the test set. CNN-100 correctly predicts 87 and 37 UAV signals, whereas XGB-100 scores 2 and 0 for medium and low SNR signals, respectively. This means CNN-100 already learned to predict medium and low SNR images, whereas XGB-100 did not. As a reminder, each category has exactly 100 recordings so the number of correct predictions is also the accuracy percentage for a given category. The full set of results is registered in Table 7, followed by an interpretation. The results for the UAV data are repeated in Figure 7 for visualization purposes, with image classifiers using dotted lines and the coefficient classifiers using solid lines.

As CNN-100 did not have medium or low SNR training images, its good performance on these test categories can only be explained by the features of CNN-100 learning to successfully predict the high SNR UAV data, features that are transferable to the low SNR categories. XGB-100, on the other hand, performs identical to CNN-100 for high SNR UAV signals, but the features it learned to predict in this category did not transfer towards allowing XGB-100 to predict low SNR UAV signals. Next, by increasing the training data, XGB-200 catches up to CNN-100 in that it can now also detect the medium and low SNR images that were missed by XGB-100, but it still cannot detect the lowest SNR category, scoring 0%. CNN-200, on the other hand, scores 80% on this category, although again, neither CNN-200 nor XGB-200 has training data from the lowest SNR category. The good performance of CNN-200 on the lowest SNR category is again understood as being the result of learning features that are more transferable. Finally, XGB-300 can indeed reach successful prediction of the lowest SNR UAV category, but only after specifically having very low SNR training data. At this point, XGB-300 and CNN-300 have the same results as were introduced prior. What needs to be emphasized is the following:XGBoost specifically needing lower SNR UAV data in its training set before it can successfully classify low SNR UAV signals. In Figure 7, this is indicated by the sudden steep declines in performance (XGB models represented with solid lines), whereas the CNN models degrade gradually.Even the well-trained XGB-300, which scores 100% on low SNR UAV signals taken in an indoor setting (Environment 1), can only score 44% and 1% on Environments 2 and 3. CNN-300, on the other hand, scores 81% and 53%. As a reminder, neither CNN-300 nor XGB-300 have seen any training data from Environments 2 or 3.

From the above, we note that, although both models suffer a degradation in new RF environments, XGB suffers to a greater extent. Based on the trends (i.e., increasing performance when increasing the training data), we can predict that XGB may eventually improve its performance in RF environments 2 and 3 if given UAV data from those specific locations. For the deployment and portability of UAV detectors, however, it is not practical to expect UAV training data from every possible deployment location, especially if using a mobile/portable UAV detector. This leads to the conclusion that image-based classifiers are more robust to changing RF conditions.

## 6. Conclusions

In this work, we have introduced a new UAV dataset that is intended to be easy to extend, not requiring any proprietary software or lab-grade equipment. The goal behind this was to lower the entry barrier for performing UAV DL/ML-based RF detection, as well as allowing for the testing of a newly introduced robustness metric. As the comparison of existing UAV detector models is largely inconclusive, having similar accuracy scores, it is necessary to determine which model can maintain its accuracy score under diverse conditions. This is only possible by extending the UAV test set and ensuring the test data are sufficiently different from the training set. In this regard, we especially focused on testing image-based classifiers against coefficient-based classifiers for low SNR UAV detection. As low SNR performance is a great concern for DL/ML-based RF classification tasks, we created a test set with multiple low SNR category test categories taken in different RF channel conditions. We found that, while all models are susceptible to performance degradation in new RF environments, image-based classifiers tend to be more robust. To investigate the reason, we performed further tests and found that image based classifiers tend to learn features that are more transferable.

There are certain limitations to this study. Firstly, only the 2.4 GHz ISM band was considered. It is not known how the different classifiers will perform in other frequency bands. Although we tested robustness with regards to low SNR conditions for different RF environments, future directions for this field could further increase the robustness, such as by introducing frequency shifting/channel hopping of the UAV signal. Some UAV models allow for the specific frequency channel to be selected. Robustness needs to be further extended to test for detecting UAV signals in frequency channels not included in the training set. Another future direction that would be beneficial in real world applications is to test the efficacy of the proposed method against UAV swarms, or against UAVs utilizing adversarial machine-learning techniques aimed at defeating DL-based classifiers. We recommend more types of robustness tests to be performed, in order to add to the real-world usability and portability of DL-based UAV RF detectors.

## Figures and Tables

**Figure 1 sensors-24-07339-f001:**
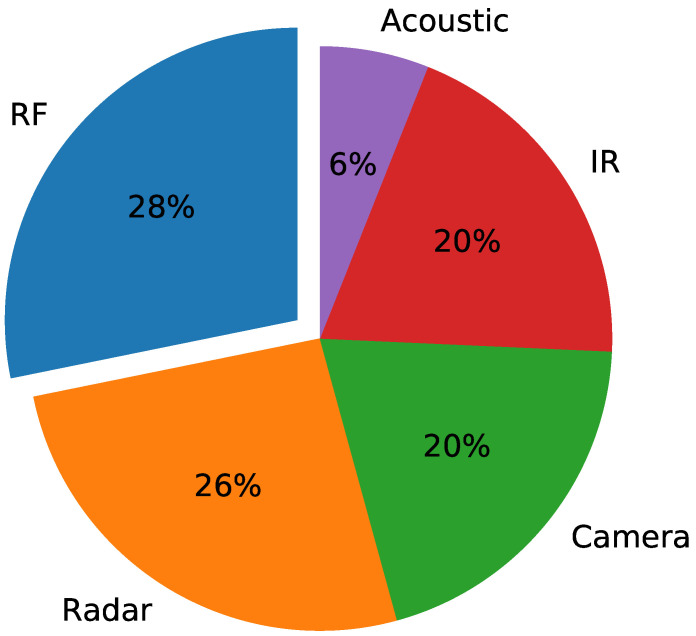
Market study breakdown of counter-UAV techniques [1].

**Figure 2 sensors-24-07339-f002:**
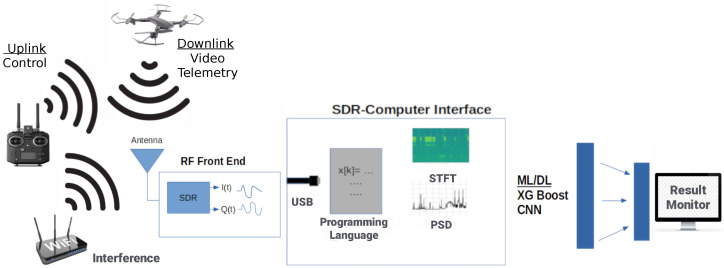
Block diagram of UAV detection via passive RF scanning and ML/DL techniques.

**Figure 3 sensors-24-07339-f003:**
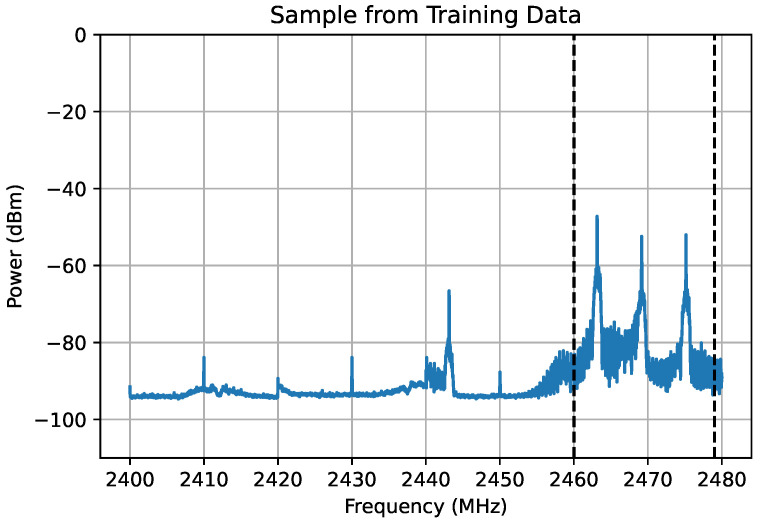
Example training data, showing the UAV control signal isolated between dotted black lines.

**Figure 4 sensors-24-07339-f004:**
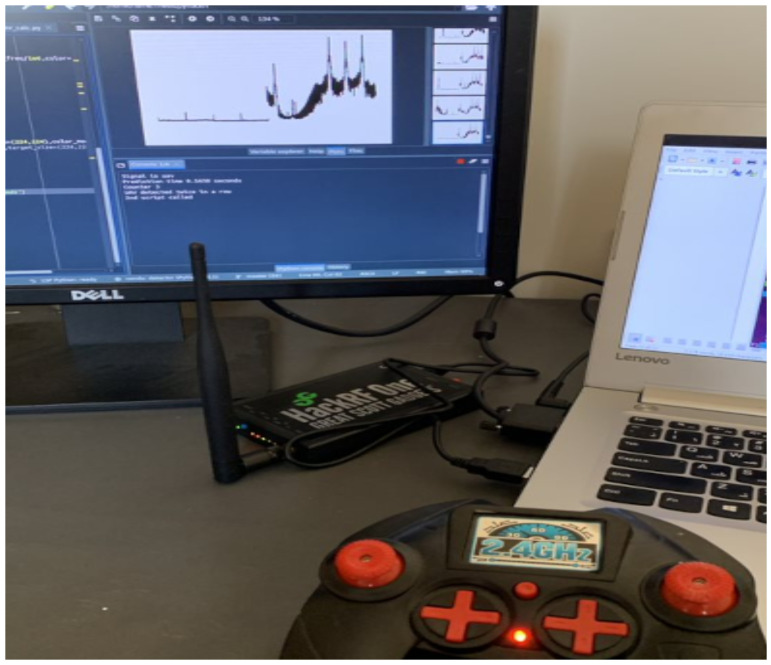
Hardware and software setup along with UAV controller.

**Figure 5 sensors-24-07339-f005:**
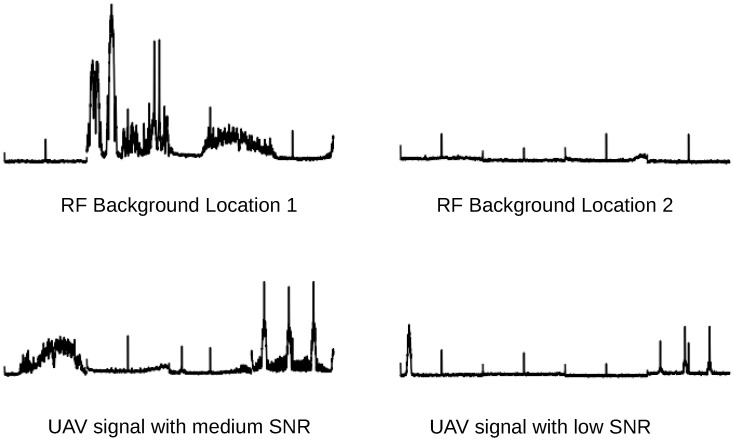
Examples from the test dataset. The UAV signal is present at the right edge of the bottom two images, showing three peaks which decay with distance.

**Figure 6 sensors-24-07339-f006:**
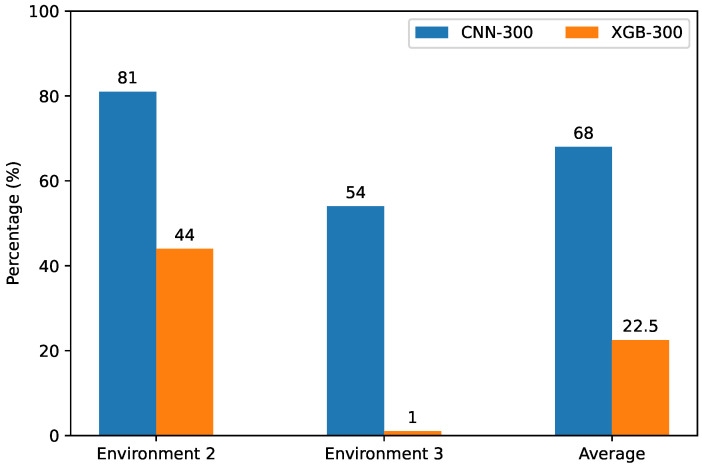
Robustness scores for low SNR performance.

**Figure 7 sensors-24-07339-f007:**
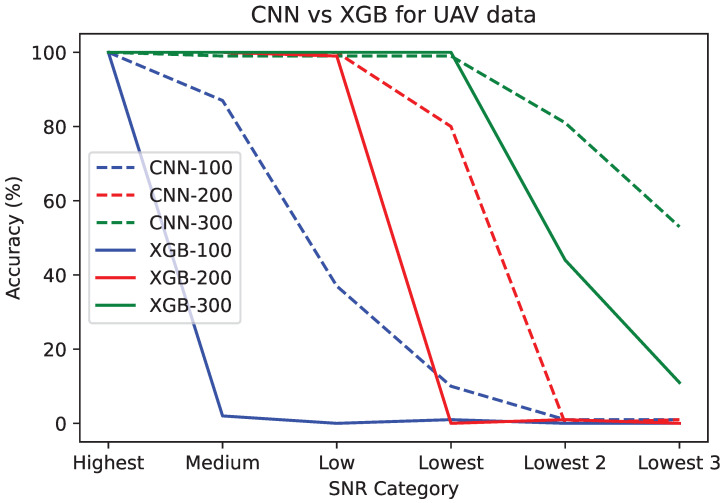
Accuracy Plots for different UAV test categories.

**Table 1 sensors-24-07339-t001:** Main neural architecture.

Layer	Size	Activation
Input	112 × 112 × 1	–
Conv2D	128 filters	ReLu
MaxPooling2D	–	–
Dropout	0.3	–
Conv2D	64 filters	ReLu
MaxPooling2D	–	–
Dropout	0.3	–
Conv2D	32 filters	ReLu
MaxPooling2D	–	–
Dropout	0.3	–
Conv2D	16 filters	ReLu
MaxPooling2D	–	–
Dropout	0.3	–
Dense	256 neurons	ReLu
Dense	2 neurons	Softmax

**Table 2 sensors-24-07339-t002:** Experimental models of fixed architecture defined by training data make-up.

Model	Training Data
CNN-100	50 RF background, 50 UAV high SNR
CNN-200	100 RF background, 50 UAV high SNR, 50 UAV medium SNR
CNN-300	100 RF background, 50 UAV high SNR, 50 UAV medium SNR, 50 images UAV low SNR, 50 images UAV lowest SNR
XGB-100	Same as CNN-100 but with 1-D arrays
XGB-200	Same as CNN-200 but with 1-D arrays
XGB-300	Same as CNN-300 but with 1-D arrays

**Table 3 sensors-24-07339-t003:** Accuracy and detection metrics for CNN-300 versus XGB-300.

Model	Precision	Sensitivity	*F*1	Accuracy
CNN-300	99.5%	99.3%	99.4%	99.2%
XGB-300	99.5%	100%	99.8%	99.7%

**Table 4 sensors-24-07339-t004:** Accuracy performance per category.

Model	Back-Ground 1	Back-Ground 2	UAV High	UAV Med	UAV Low	UAV Lowest
CNN-300	98	100	100	99	99	99
XGB-300	98	100	100	100	100	100

**Table 5 sensors-24-07339-t005:** Accuracy Performance on “Lowest SNR” category for three RF environments.

Model	UAV Lowest Environment 1	UAV Lowest Environment 2	UAV Lowest Environment 3	Degradation Environment 2	Degradation Environment 3
CNN-300	99	81	53	−18	−46
XGB-300	100	44	1	−56	−99

**Table 6 sensors-24-07339-t006:** Robustness scores.

Model	Robustness 2	Robustness 3	Average Robustness
CNN-300	81%	54%	68%
XGB-300	44%	1%	22.5%

**Table 7 sensors-24-07339-t007:** Results comparing the performance of image vs coefficient classifiers on the fixed test set.

Model	Back-Ground 1	Back-Ground 2	UAV High	UAV Med	UAV Low	UAV Lowest	UAV Environment 2	UAV Environment 3	Overall Accuracy
CNN-100	95	100	100	87	37	10	1	1	53.9%
XGB-100	99	100	100	2	0	1	0	0	37.8%
CNN-200	98	100	100	99	100	80	0	1	72.3%
XGB-200	100	100	100	100	99	0	1	0	62.5%
CNN-300	98	100	100	99	99	99	81	53	91.3%
XGB-300	98	100	100	100	100	100	44	1	80.4%

## Data Availability

The dataset used will be made available via a link after publication.

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
