# Peer review of "Robustness of Deep-Learning-Based RF UAV Detectors"

_sensors, 2024, doi:10.3390/s24227339_

Round 1
Reviewer 1 Report
Comments and Suggestions for Authors
This paper studies the robustness of deep learning based RF UAV detectors.
(1) Equation (4) should be further stated with proper reference.
(2) Why do you consider the "Baseline as CNN-300 and XGB-300"? Other performance should be given.
(3) Do you consider the flying mode? Does the UAV' moving have effects on your method?
Comments on the Quality of English LanguageNo
Reviewer 2 Report
Comments and Suggestions for Authors
The paper introduces a novel dataset and explores the robustness of UAV detection using RF signals and machine learning techniques. The authors compare different ML/DL models, emphasizing the importance of robustness under varying RF conditions.
1:The abstract lacks specificity and fails to provide a concise summary of the study's major contributions and results. It mentions the general problem of UAV detection and the use of machine learning techniques but does not clearly articulate the unique aspects of the proposed approach or the specific findings regarding robustness. This makes it difficult for readers to quickly grasp the significance of the work.
2:The introduction section does not sufficiently cover the existing state of research in RF-based UAV detection. It briefly mentions the general trend towards using machine learning and deep learning for UAV detection but lacks a detailed comparison of various existing methods, their strengths, and their weaknesses. Moreover, the introduction does not clearly define the gap in the current research that this paper aims to address, nor does it articulate the importance of robustness in UAV detection.
3:The literature review section is somewhat superficial and does not provide a comprehensive overview of the relevant research. It cites a limited number of studies and does not delve deeply into the methodologies and results of these studies. There is a lack of critical analysis of existing approaches, and the review does not adequately highlight the need for a focus on robustness in UAV detection methods.
4:The literature review fails to establish a strong theoretical framework for the study. It does not connect the reviewed literature to the theoretical underpinnings of machine learning and signal processing techniques used in UAV detection. This omission makes it difficult to understand how the proposed research builds on and contributes to existing knowledge.
5:The paper would benefit from including the reference "An advanced scheme for range ambiguity suppression of spaceborne SAR based on blind source separation." “Micro-UAV detection and classification from RF fingerprints using machine learning techniques” This reference could provide valuable insights into advanced signal processing techniques that could enhance the robustness of UAV detection systems. Including this reference would help to situate the study within a broader context of signal processing research and highlight potential avenues for improving the proposed methods.
6:The experimental design lacks clarity and detail. The paper does not thoroughly explain how the experiments were conducted, including the specific configurations of the hardware and software used, the number of samples collected, and the specific settings for the machine learning models. This makes it challenging to replicate the study or verify its findings.
7:The methodology for testing robustness is not well-defined. The paper discusses the importance of robustness but does not provide a clear and systematic approach to measure and compare the robustness of different classifiers. There is no detailed explanation of the criteria used to evaluate robustness or how different environmental conditions and SNR levels were simulated and controlled.
8:The presentation of the results is not sufficiently detailed. The paper provides accuracy metrics for the classifiers but does not offer a comprehensive analysis of the results. There is a lack of visual aids such as graphs and charts that could help illustrate the performance differences between the classifiers under various conditions.
9:The results section does not include a rigorous statistical analysis. There is no discussion of the statistical significance of the findings or any confidence intervals for the reported metrics. This makes it difficult to determine whether the observed differences in classifier performance are meaningful or could be due to random variation.
10:The paper does not adequately address the limitations of the study. There is no discussion of the potential weaknesses in the experimental design, the dataset, or the analysis methods. Additionally, the suggestions for future work are vague and do not provide a clear direction for further research.
Comments on the Quality of English Language1: Several sentences in the manuscript are ambiguous and could be interpreted in multiple ways.
Location: Abstract (Page 1), Introduction (Page 1-2), Methodology (Page 6).
Example: "A new data set is introduced, with training and test data taken in different channel conditions." This sentence is vague and does not clearly explain how the dataset was created or the specific conditions considered.
2: There are several grammatical errors throughout the manuscript.
Location: Introduction (Page 1), Methodology (Page 6), Results (Page 10).
Example: "The results does not hold when replacing UAV test categories with UAV recordings from different RF channel conditions." This should be corrected to "The results do not hold when replacing UAV test categories with UAV recordings from different RF channel conditions."
3: The manuscript contains several run-on sentences that hinder readability.
Location: Introduction (Page 1-2), Methodology (Page 6-7).
Example: "The setup to collect the data is shown in Figure 4. The SDR was connected to a Lenovo Laptop with an Intel Core i7-7500U CPU @ 2.70 GHz, 8 GB of RAM, and running an Ubuntu 20.4 operating system. Python scripts were developed to access the SDR, tune it to the desired center frequency, stream IQ samples, and then tune to the next center frequency, until the four consecutive scans were complete." This could be broken down into shorter, clearer sentences.
Reviewer 3 Report
Comments and Suggestions for Authors
I think it is a valuable feature to be able to effectively perform RF detection by UAVs. Furthermore, I think it is an important validation that it is validated using existing equipment, CNN and XGB. This is a study that will be very good if it is further validated for applicability and linked to the development of new equipment.
Please consider adding the following
+What is the rationale for requiring such high accuracy for the purpose of flying UAVs? What is the required value?
+What is the actual timing of the experiment? What is the target area? How large is the area?
Reviewer 4 Report
Comments and Suggestions for Authors
Now all the necessary changes are done. Only describe some more about future works in Conclusions.
Author Response
Comment 1: Now all the necessary changes are done. Only describe some more about future works in Conclusions.
Response 1: Thank you so much for your time and reviewing our manuscript. With regards to describing some more future works in the conclusion, we have added some new future directions in Page 16, Last Paragraph, Lines 577-579.
Reviewer 5 Report
Comments and Suggestions for Authors
In this work, the authors prepare a new dataset designed to test for robustness given the fact that existing datasets are biased as they test only some specific situation and ignore the other eventhough these ignoded scienarios occur frequently during UAV operation. In contrast of existing datasets for UAV training, they do not reserve the test samples from the training dataset. Instead they perform real experiments and feed the model the test the robustnes of the training. The also propose to allow for multiple categories of test data based on channel conditions. This kind of paper regarding the proposal of robust and complete datasets is always welcome in the AI community.
The paper is well-written and has been ammended correctly to imporve its contents. I suggest to accept it in its present form.
Author Response
Dear Reviewer 5,
Thanks a lot for spending time to review our paper.
Round 2
Reviewer 1 Report
Comments and Suggestions for Authors
This paper introduces a new dataset specifically designed to test for robustness.
(1) The advantages of the RF Detectors should be discussed. In addition, the advantages and disadvantages of other detectors in Fig. 1 should also be discussed.
(2) What targets do you detect? Different targets may have different detection performance.
(3) In practical applications, since the detection targets usually be obstacles and neighboring UAVs, the safety control of UAV can be achieved by the detection results. As a result, you should discuss the future applicaiton of the methods in you paper, such as "Secure Finite-Horizon Consensus Control of Multiagent Systems Against Cyber Attacks", "Secure Cooperative Guidance Strategy for Multi-Missile System With Collision Avoidance", "Model Predictive Formation Tracking-Containment Control for Multi-UAVs With Obstacle Avoidance".
Comments on the Quality of English Languageno
Reviewer 2 Report
Comments and Suggestions for Authors
While the authors have made several revisions in response to my initial comments, some key concerns remain insufficiently addressed. The manuscript has shown some improvement in certain sections, but there are still significant gaps that need attention, particularly in terms of specificity, clarity, and completeness.
1. The revised abstract is still lacking in specificity. While the authors have removed some general background information, the abstract does not clearly articulate the study's unique contributions or the main results. It remains too vague for readers to quickly grasp the importance and novelty of the work. The abstract should explicitly state the specific problem addressed, the methods used, the unique contributions of the study, and the key results. Include quantitative results or specific metrics to make the contributions more concrete.
2. Although the authors have expanded the introduction, it still does not sufficiently define the gap in the existing research or the importance of robustness in UAV detection. The discussion of existing methods remains somewhat superficial and lacks a critical analysis that clearly situates the current study within the broader context of UAV detection research.
3. The revised literature review is still somewhat limited in scope and depth. While it now includes additional references and discussions, it lacks a comprehensive and critical analysis of the reviewed works. The theoretical framework connecting machine learning and signal processing techniques to UAV detection is still not sufficiently developed.
4. The authors have added some details about the experimental setup, hardware configurations, and data collection methods. However, the explanation remains somewhat fragmented and lacks coherence. Additionally, the discussion on the robustness testing methodology is still not sufficiently clear or systematic. Provide a more structured and detailed description of the experimental design, including the specific configurations, parameters, and settings used. Clearly define the criteria for robustness evaluation and describe in detail how different environmental conditions and SNR levels were simulated and controlled. Consider using diagrams or flowcharts to visually represent the experimental setup and procedure.
5:While some language issues have been corrected, the manuscript still contains numerous grammatical errors, ambiguous sentences, and run-on sentences that hinder readability. Conduct a thorough language review to correct grammatical errors, improve sentence structure, and ensure clarity and conciseness. Consider consulting a native English speaker or a professional editor to refine the language.
